# A global analysis of coral bleaching over the past two decades

S. Sully [1], D.E. Burkepile [2,3], M.K. Donovan [3], G. Hodgson [4] & R. van Woesik [1]

Thermal-stress events associated with climate change cause coral bleaching and mortality that threatens coral reefs globally. Yet coral bleaching patterns vary spatially and temporally. Here we synthesize field observations of coral bleaching at 3351 sites in 81 countries from 1998 to 2017 and use a suite of environmental covariates and temperature metrics to analyze bleaching patterns. Coral bleaching was most common in localities experiencing high intensity and high frequency thermal-stress anomalies. However, coral bleaching was significantly less common in localities with a high variance in sea-surface temperature (SST) anomalies. Geographically, the highest probability of coral bleaching occurred at tropical mid-latitude sites (15–20 degrees north and south of the Equator), despite similar thermal stress levels at equatorial sites. In the last decade, the onset of coral bleaching has occurred at significantly higher SSTs (~0.5 °C) than in the previous decade, suggesting that thermally susceptible genotypes may have declined and/or adapted such that the remaining coral populations now have a higher thermal threshold for bleaching.

[1] Institute for Global Ecology, Florida Institute of Technology, 150 West University Blvd., Melbourne, FL 32901, USA. [2] Department of Ecology, Evolution, and Marine Biology, University of California, Santa Barbara, CA 93106, USA. [3] Marine Science Institute, University of California, Santa Barbara, CA 93106, USA. [4] Reef Check Foundation 13723 Fiji Way, B-2 Marina del Rey, CA 90292, USA. Correspondence and requests for materials should be addressed to R.v.W. (email: rvw@fit.edu)

Coral reefs are the world's most diverse marine ecosystems. They provide billions of dollars in economic value through coastal protection, food, tourism, and pharmaceuticals from the sea[1]. Rapid increases in sea surface temperatures (SSTs) are increasing the frequency and intensity of coral bleaching events[2–6], during which corals lose their endosymbiotic algae — a primary energy source for most reef corals. Coral bleaching can cause coral morbidity and mortality, which leads to losses of coral cover, dramatic changes to coral community composition, and even rapid reorganization of coral-reef-fish communities[7,8].

Most studies that examine coral response to coarse-grained global atmospheric-ocean circulation models predict that within the next 80 years few coral reefs will survive in tropical oceans[9]. The 2014–2017 global coral-bleaching event, the third in the last 20 years, killed corals and other reef organisms over thousands of square kilometers[8,10]. Yet, both satellite data and local field studies show that not all coral reefs are equally exposed to severe temperature stress events[10]. Even where they are, corals show local and regional variation and species-specific responses to thermal stress[11–14]. Together, these studies show that the relationship between anomalously high SSTs and coral bleaching varies over space and time. Compared with coarse-grained global models that predict minimal coral survival in the tropical oceans within the next 100 years, recent field work shows considerable geographic variability in both temperature stress and coral survival[11–14]. This mismatch between global models and field results underscores the urgent need to develop better models that accurately predict the geographical heterogeneity of coral bleaching as corals respond to ocean warming.

We take a spatially explicit approach to examine the response of coral communities to thermal stress events at 3351 sites in 81 countries (Fig. 1 and Supplementary Figs. 1 and 2). Importantly, the coral community bleaching response was recorded using the same standardized protocol at each site across a suite of changing environmental variables from 1998 to 2017. The environmental variables encompassed several high thermal-stress events, including El Niño conditions, during which large parts of the tropical oceans were warmer than usual — increasing the probablility of coral bleaching. Degree Heating Weeks (DHW) has become a standard global predictor of bleaching[15], with 1 DHW defined as 1 °C above the long-term climatology for the warmest month at a given locality. Severe bleaching is common at 8 DHW and above[16]. We construct a generalized linear mixed model in a Bayesian framework to predict the probability of coral bleaching by including DHW and other temperature metrics (Supplementary Table 1), latitude, depth, and coral diversity. Our aim is to improve coral bleaching predictions and obtain a more comprehensive understanding of geographic differences in the coral response to thermal stress. We show that coral bleaching was more prevalent in localities with high SST, both in absolute degrees and in DHW, and in localities with frequently high SST anomalies. Coral bleaching was also higher in areas with high rates of change in SST but lower in areas with high variability in SST. Bleaching probability was highest at mid-latitude sites despite equivalent thermal stress at equatorial sites.

## Results and Discussion

Significantly more coral bleaching occurred at mid-tropical latitude sites, between 15 and 20° north and south of the Equator than in the equatorial regions, where coral diversity is highest (Fig. 2, Supplementary Figs. 3 and 4, & Supplementary Table 2). Notably there was no correlation (Spearman's $rho = 0.313$, $p$-value = 0.297) between bleaching prevalence and the number of study sites. The clustering of coral bleaching at 15–20° north and south of the Equator was not, however, a consequence of higher thermal anomalies at those latitudes than elsewhere (Supplementary Figs. 5–15). Our finding of less coral bleaching in equatorial regions, where coral diversity is the highest on a global scale, contrasts with other studies at the regional scale, which showed that the most extensive bleaching occurred at the most diverse reefs in the Commonwealth of the Northern Mariana Islands[17]. Unless there was less thermal stress in the low-latitude tropics than elsewhere, which we did not detect in this study, our results lead to several hypotheses that potentially explain differential coral bleaching among latitudes. We hypothesize that the low-latitude tropics bleached less because: (i) of the geographical differences in species composition, (ii) of the higher genotypic diversity at low latitudes, which include genotypes less susceptible to thermal stress, and (iii) some corals were preadapted to thermal stress because of consistently warmer temperatures at low latitude prior to thermal stress events. These hypotheses are not mutually exclusive and several of these mechanisms could be operating in concert, resulting in less coral bleaching at low latitudes.

Coral bleaching was also significantly lower in localities with a high variance in temperature anomalies, taken over weekly intervals (Fig. 2 & Supplementary Figs. 16–18). We also note that coral bleaching was negatively related to the standard deviation of thermal stress events (Fig. 2). We found that the global correlation between lower coral bleaching and higher SST variance, at weekly scales, corroborates previous regional studies that showed a small daily temperature range was consistently the best metric for predicting bleaching prevalence, with greater SST variability reducing the odds of coral bleaching[3,12–14]. Our results suggest that localities that commonly experience large daily, weekly, or seasonal SST ranges may harbor corals, and strains of coral symbionts, that are more resistant to SST extremes[18]. Further research should untangle this spatial heterogeneity in SST

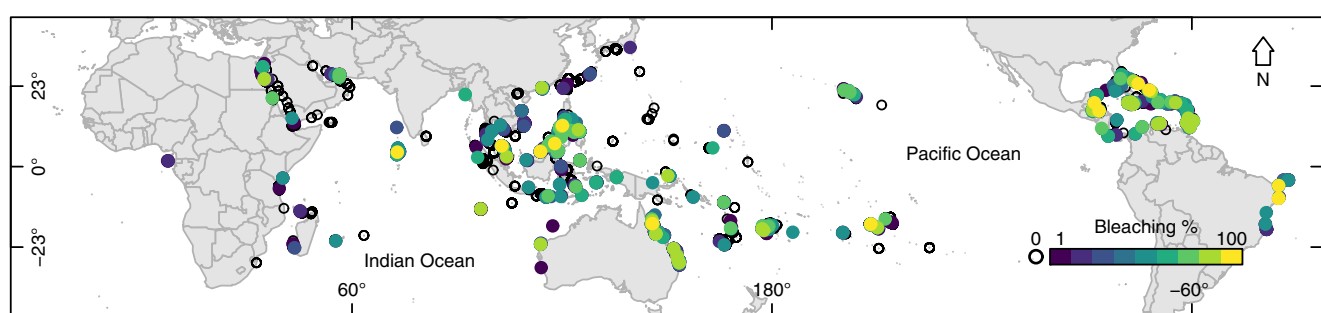

**Fig. 1** Coral bleaching distribution. Prevalence of coral bleaching presented as a percentage of the coral assemblage that bleached at survey, measured at 3351 sites in 81 countries, from 1998 to 2017. White circles indicate no bleaching. Colored circles indicate 1% bleaching (blue) through 100% bleaching (yellow)

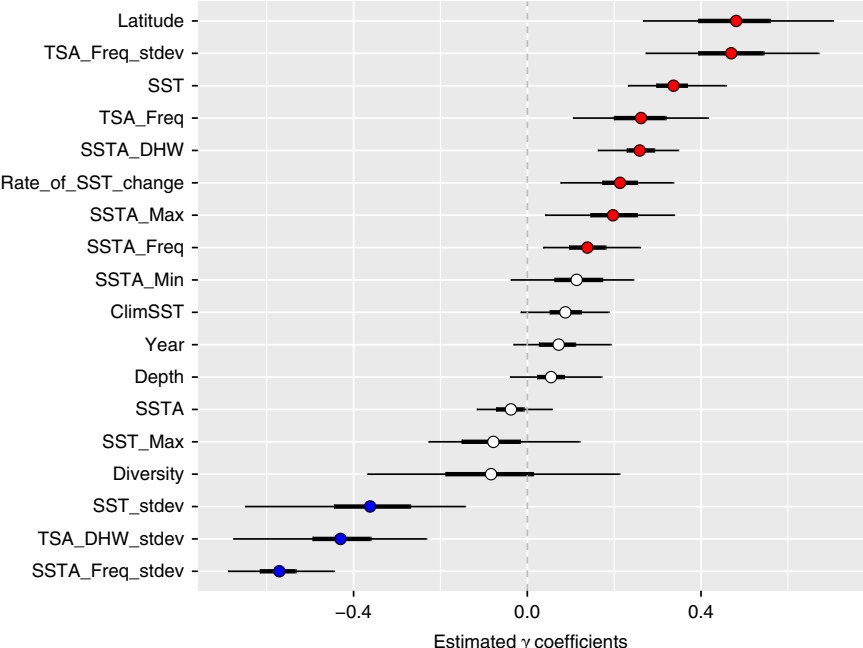

**Fig. 2** Model parameter coefficients. Relationship between the percentage of coral colonies bleached and environmental variables across all depths within a Bayesian framework with mean values (circles) and 95% credible intervals (the thin black horizontal lines) as well as 50% credible intervals (the thick black horizontal lines) at 3351 sites in 81 countries, from 1998–2017 (all definitions are outlined in detail in Supplementary Table 1). Latitude is the number of degrees north or south of the equator at which the survey occurred. SST is the sea surface temperature during the field survey period. Freq is frequency. Min is minimum. Max is maximum. Stdev is standard deviation. DHW is degree heating weeks. A is anomaly. TS is thermal stress. Diversity is the number of species confirmed present in the ecoregion in which each survey was conducted. Rate_of_SST_change is the annual rate of SST change from 1984 to 2017 at a 1 km resolution. Year is the years of survey. Clim is climatological. Depth is the depth in meters. Red dots show a positive contribution to bleaching likelihood, blue dots show a negative contribution to the likelihood of coral bleaching, and white dots show no significant contribution to bleaching likelihood (95% credible interval crosses 0)

variance and determine to what extent acclimation versus adaptation is contributing to reduced coral bleaching prevalence.

The results that coral bleaching was less common in the equatorial regions, with high coral diversity[19], agree with paleoecological studies that show greatest stability and lowest extinction in the tropics through rapid climate change[20]. Furthermore, recent studies show that marine taxa track climate velocity[21], which is the rate and direction that the climate shifts across the seascape. The predicted climate velocities in the oceans show that the lowest variance in species-range shifts are occurring within ten degrees latitude of the Equator[22]. Although the tropics may be potentially more stable through climate changes than elsewhere, several modeling studies have nevertheless predicted high species loss near the Equator with increasing temperatures[22,23]. Yet, to make such predictions, these modeling studies only consider the narrow thermal range of modern equatorial organisms, and do not consider the role of species or genotypic diversity in driving the differences in thermal responses, or the potential of the genetic standing stock to adapt to thermal stress.

In concordance with the global predictions[24–26], in the last decade, coral bleaching has increased in frequency and intensity (Fig. 3). Yet, in the last decade, the onset of coral bleaching has occurred at significantly higher SSTs (~0.5 °C) than in the previous decade (Fig. 4). At the thousands of sites surveyed, the mean SST recorded during coral bleaching in the first decade of the dataset, from 1998 to 2006, was 28.1 °C, whereas the mean SST recorded during coral bleaching in the second decade, from 2007 to 2017, was 28.7 °C. This change in coral-bleaching temperature is significantly different (Likelihood ratio test, $Pr(>\chi^2) = 0.001$) between decades (Fig. 4). The increase in over half a degree

celsius in coral-bleaching temperature suggests that past bleaching events may have culled the thermally susceptible individuals, resulting in a recent adjustment of the remaining coral populations to higher thresholds of bleaching temperatures[26–28] (Supplementary Figure 19). Coral communities also may have acclimatized to increasing SSTs, highlighting the need for further research to understand the context dependencies of this trend towards a greater temperature threshold.

Our model showed that rates of change in SST are strong predictors of coral bleaching with faster rates of change correlating with higher levels of bleaching (Fig. 2 and Supplementary Figure 20). Global models predict a mean increase in SST of 0.027 °C per year from 1990 to 2090[29], which is almost double the rate (0.015 °C per year) of the previous 30 years. As SSTs continue to increase more rapidly, more localities are likely to experience coral bleaching. We show that coral bleaching is predictable, at large scales, by the intensity and the variance in frequency of extreme, high-SST events. We demonstrated that equatorial areas and areas with greater exposure to short-term SST fluctuations may be more resilient to high temperature events, and therefore may be important targets for conservation given their increased likelihood of persisting into the future[30]. Coral bleaching has had unprecedented negative effects on coral populations worldwide, and immediate action globally to reduce carbon emissions is necessary to avoid further declines of coral reefs.

## Methods
**Biological and environmental data**. The coral bleaching data were composed of the Reef Check dataset (reefcheck.org), collected by a mixture of professional scientists (56%) and trained and certified citizen-scientists (44%) using a standardized transect protocol[31]. The Reef Check data are the only field-based coral-

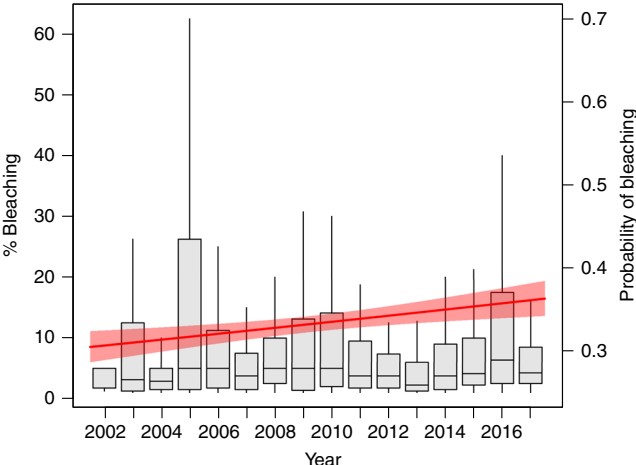

**Fig. 3** Percent and probability of coral bleaching over time. Percent of coral bleaching and probability of coral bleaching measured at 3351 sites in 81 countries, from 2002 to 2017. The boxplots are of the percent coral bleaching, which is measured on the left y axis. The center line is the mean percent bleaching, the bounds of the boxes are the interquartile range (25 and 75%), and the whiskers are the 95% range. The red line is the probability of coral bleaching over time, measured on the right y axis, and the shaded red region is the 95% confidence interval

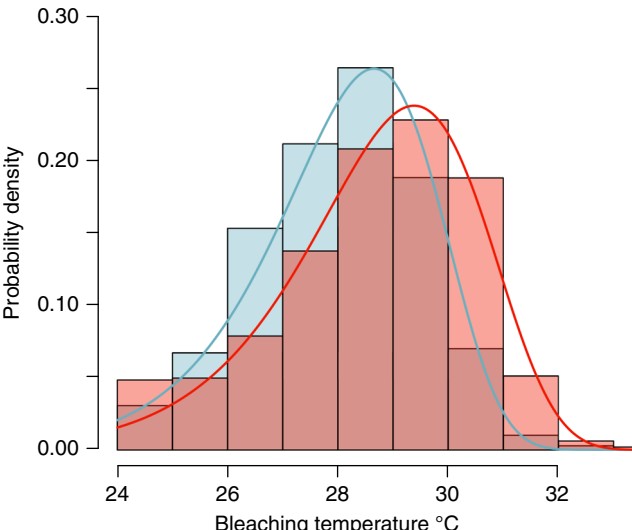

**Fig. 4** Probability density distributions of coral bleaching. Probability density distributions of coral bleaching from 1998 to 2006 (blue shade) and from 2007 to 2017 (peach shade), the mauve shade is where the distributions overlap; the blue and red lines show the best-fit Weibull probability density distributions (for the 1998 to 2006 data, the Weibull shape is 18.895 and the scale is 28.622, whereas for the 2007 to 2017 data the Weibull shape is 19.346, and the shape is 29.413). The change in coral-bleaching-sea-surface temperature is significantly different (Likelihood ratio test, $\text{Pr}(>\chi^2) = 0.001$) between decades

reef data collected on a global scale using a standardized methodology and have been used in numerous global and regional analyses[31,32]. The validity of Reef Check data has been well documented[32]. The data comprised 9215 data points, for 3351 sites (Supplementary Figs. 1 and 2 & Supplementary Table 3), from 81 countries, collected from 1998 to 2017. The mean frequency for field sampling was 2.75 (standard deviation = 3.17) times over the sampling period (see supplementary document for more details on sampling effort). In addition to a suite of temperature metrics, ecological data, and coral diversity data obtained from J.E.N. Veron[19] (Supplementary Figure 21 & Supplementary Table 1), the dataset includes

counts of the number of coral colonies showing bleaching (i.e., the percent of reef corals that were recorded as bleached), which was classified as site-wide bleaching. There was also a categorized estimate of the percentage of each coral colony that was bleached (i.e., per colony bleaching) at each site during each sampling period. Here we used the data pertaining to the site-wide bleaching, which was expressed as a percentage. We also examined the prevalence of coral bleaching per coral ecoregion (as defined by Veron et al. 2015)[19]. We used the global Coral Reef Temperature Anomaly Database (CoRTAD Version 6) from the National Oceanic and Atmospheric Administration (www.nodc.noaa.gov/sog/cortad/) to predict coral bleaching prevalence and intensity across reefs worldwide. All CoRTAD variables were weekly data provided on a grid cell basis, of ~4 km resolution, from 1982 to 2017 (Supplementary Table 1).

**Data analysis.** The covariates that we used in the analysis are summarized in Supplementary Table 1; a Pair-wise Pearson's correlation of coefficients was used to determine which covariates were highly collinear (Supplementary Fig. 22). We conservatively discarded 14 predictor variables whose correlation coefficients were >0.65 with co-occurring predictors. One-hundred and fifty-three sites (4%) were removed that had missing data for the environmental variables or fell outside of ecoregion boundaries. We used generalized linear mixed models, within a Bayesian framework, to examine the influence of the covariates on coral bleaching. We standardized each covariate to improve the stability of our model. Some sites were repeatedly surveyed and therefore site was treated as a random effect. Coral bleaching for a given observation ($o_i$) was assumed to follow a series of Bernoulli processes ($p_i$) captured as a negative binomial distribution[33] using a log-link function, since the data were zero-inflated,

$$o_i \sim \text{negative binomial}\,(p_i, k), \qquad (1)$$

$$\text{Expected}\,(o_i) = p_i, \qquad (2)$$

$$\text{Variance}\,(o_i) = p_i + p_i^2/k, \qquad (3)$$

$$\log(p_i) = b_0 + \gamma_1\Big(\big(\text{covariate}_{i,1} - \text{mean covariate}_1\big)/\text{sigma covariate}_1\Big)$$
$$+\dots\ \gamma_n x_{i,n} + a_{i,s}, \qquad (4)$$

$$a_s \sim \text{norm}(R_r, \tau), \qquad (5)$$

$$R_r \sim \text{norm}\,(g_r, \text{T}), \qquad (6)$$

$$g_r = \mu + b_{\text{div}}d_r, \qquad (7)$$

where $b_0$ is the intercept, $\gamma$ are coefficients, $x$ are environmental covariates, $a$ are random effects of site ($s$), which hierarchically follow a normal distribution (norm) from the random effect ($R$) of ecoregion ($r$) with mean $g_r$, $b_{\text{div}}$ is the coefficient for diversity ($d_r$) introduced at the ecoregion level, $\mu$ is the overall mean, and $\tau$ and T are variance across site and ecoregion, respectively. Covariates were modeled with flat normal priors. The Bayesian model was implemented in R[34] and run through the *rjags* package that calls JAGS[35], with 3 chains, a burn-in of 4000, and 5000 iterations. The trace plots were examined for convergence, and posterior predictions were compared with simulated values from the same model[36].

Posterior predictive checks were used to assess evidence of lack of fit between model estimates and data. A Bayesian P-value based on the mean was ill-suited for the zero-inflated model, therefore we examined the fit to the mean for only non-zero bleaching values, and obtained a P-value of 0.503. A separate posterior check was undertaken for the zero bleaching values, to compare simulated data and observed zero bleaching, which indicated that the simulated data correctly estimated zero coral bleaching 50% of the time, and 3.4% (standard deviation 4.4%) coral bleaching when the simulated data was an overestimate.

To spatially examine the environmental variables that potentially impact coral bleaching, we determined the mean value of each variable whose credible intervals did not cross zero (Fig. 2), per ecoregion. The value of the variable in an ecoregion is then reported as the number of standard deviations from the variable's mean over all ecoregions. To be included in this analysis, an ecoregion was required to have had at least 10 surveys over the 1998 to 2017 sampling period.

**Reporting Summary.** Further information on experimental design is available in the Nature Research Reporting Summary linked to this article.

## Data Availability

All the R code, Reef Check data, and diversity data for the analysis are available at the GitHub repository for the Institute for Global Ecology https://github.com/InstituteForGlobalEcology/Coral-bleaching-a-global-analysis-of-the-past-two-decades. All Coral Reef Temperature Anomaly Database (CoRTAD) data used in this analysis are publicly available at NOAA's National Centers for Environmental Information (NCEI) webpage (https://data.nodc.noaa.gov/cortad/Version6/). All sea surface temperature (SST) data used to determine the rate of SST change are publicly available in a downloadable file titled sst.mnmean.nc at NOAA's Earth Systems Research Laboratory

(ESRL), Physical Sciences Division (PSD) webpage (https://www.esrl.noaa.gov/psd/data/gridded/data.noaa.oisst.v2.html).

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

## Acknowledgements

The environmental data were provided by the National Oceanic and Atmospheric Administration (NOAA) National Centers for Environmental Information (NCEI) and were supported in part by a grant from the NOAA Coral Reef Conservation Program (CRCP). We thank Jenny Mihaly and the thousands of volunteer scientists and citizen scientists who have collected Reef Check data since 1997. The CoRTAD data were provided by GHRSST and the US National Centers for Environmental Information, which was supported in part by a grant from the NOAA Climate Data Record (CDR) Program for satellites. NOAA_OI_SST_V2 data was provided by the NOAA/OAR/ESRL PSD, Boulder, Colorado, USA, from their Web site at https://www.esrl.noaa.gov/psd/. We would also like to thank Sandra van Woesik and the three reviewers for comments and suggestions that improved the manuscript, and the National Science Foundation (OCE 1657633 and OCE 1829393) and the Zegar Family Foundation for supporting our research. We also thank Chelsey Kratochwill for tireless assistance with the database. This is contribution number 196 from the Institute of Global Ecology at the Florida Institute of Technology.

## Author contributions

G.H. provided the data; S.S., M.D. and R.vW. developed the model and wrote the R code, R.vW. and D.B. initiated project and secured funding; R.vW. and S.S. wrote the first draft; and S.S., D.B., M.D., G.H. and R.vW. interpreted results and edited the manuscript.

## Additional information

**Competing interests:** The authors declare no competing interests.

