## [Peer Review File · Nature Communications]

Reviewers' Comments:

Reviewer #1:

Remarks to the Author:

This paper analyses a strong and consistent dataset to make a clear point that bleaching levels are higher at mid-latitudes for coral reefs, and that the threshold for bleaching is significantly higher in the 2nd decade than the first of data collection. These are two very clear and important findings and should be published – they contribute to further science on understanding global change impacts, and help inform long term management planning.

But the presentation of these results is muddled by mixing in two untested components – the relationship of these patterns with diversity of corals (no data presented) and higher reports of bleaching in the northern hemisphere (without addressing sampling artefacts that may influence this). I would have recommended minor revision to either remove or strengthen the references to diversity, but find after rereading that it is so mixed into the findings that the paper really needs a significant rewrite to do this well. Dealing with the geographic spread of sampling is also a significant addition to the conclusions from a dataset like this in a high-impact journal (and may, in fact, change the latitudinal pattern in some way). I believe resubmission is warranted, in a manuscript that focuses on the two defensible findings.

Detailed comments

42-43; 87-8- the paper does not address diversity, so it should not be mentioned, as is just correlative. It could easily – since species diversity for each of the Veron ecoregions is easily obtained. But if this is not added in, its more of a 'messaging' and subjective piece that is not backed up by any of the analysis in the paper.

80 – this is six localities, not five

90-92- this feels a bit like cherry-picking a single reference, whereas there are many studies of regional patterns on bleaching in relation to latitude and diversity

93-94 – or that some other factor is related to lower bleaching at equatorial latitudes, that is unrelated to diversity. The paper does not really address diversity, and while the higher equatorial species diversity is implicit and generally known, it not a given that higher genotypic diversity is correlated with species diversity.

111-112 "The results that coral bleaching was less common in the equatorial regions with highest diversity" ... equatorial regions have properties other than high diversity, such as the temperature gradients (velocities?) being lower than other regions. So the a priori association of genetic diversity as a reason explaining the patterns presented here is an assumption of the paper, and is not tested. It should not be presented as a primary finding. Correlation, as implied here (and it is not even demonstrated statistically), does not mean causation.

111-115 – latitudinal patterns of species diversity in corals are attributed to a wide range of Cenozoic influences (see Renema and others), rather than more recent glacial cycles, so even this citation of (terrestrial) diversity is not particularly accurate or relevant. There will be many citations for marine diversity patterns that would be more relevant, but may not indicate the same pattern.

121-123 – "do not consider the role of species or genotypic diversity in driving the differences in thermal responses, or the potential of the genetic standing stock of corals to adapt to thermal stress"

.... The role of these in the paper is speculation, but is presented as causation.

125 – yes, 'less predictable', but nevertheless highly expected for 15-20 years now, so citations to this

effect must be made, and include Hoegh-Gulberg 1999, Sheppard 2003, Hughes et al 2003 ...
132-133 – and/or lead to acclimatization of surviving corals
131-139 – this discussion on rates of change of SST should cite Donner 2009 (PLOS ONE) , who has estimated expected rates of change of temperature against adaptive capacity in coral communities. Reference #22 for this section seems a very narrow one to use, where more comprehensive syntheses have been made.
140-144 These sentences and logic are somewhat jumbled, flipping between explanations based on the study “results have important implications for improving predictions of future bleaching” and findings “We demonstrated that equatorial areas ...”, but without attributing the original ideas where they are due. This study confirms what is known about this topic with a new analysis of a complete/updated dataset. It adds strength to one hypothesis, that survival is better at equatorial latitudes because rates of warming are less (compared to an alternate hypothesis that survival might be less at the equator as absolute temperatures will be higher) but it does not tell us anything we did not already know (and in fact, this alternate hypothesis is not stated explicitly). For example, the predictions of greater survival of corals near the equator is consistent with Beyer et al. 2018 (though this paper does not make a specific statement about this), but this is not mentioned or cited.
170 – SST data only to 2012? What about later years?
Figs S1 – the predominance of bleaching in northern latitudes may be an artefact of sampling effort at these latitudes, and less in the south. To really say bleaching is more prevalent in the north sampling effort in relation to reef area needs to be analyzed. It is also possible that the higher sampling levels in the north are strongly biased to a small number of popular locations for this method, so patterns may be an artefact of this.
Methods – need to state more clearly which variates were obtained from field data, and which from the lat-long points of field sites extracted from CoRTAD data.

Reviewer #2:

Remarks to the Author:

Review of Sully et al for NC: Coral bleaching: a global analysis of the past two decades

Overall, I think this is an excellent, important, and timely paper/analysis. I am familiar with both of the core databases (ReefCheck and CoRTAD) and have published papers based on both. Although the ReefCheck data is collected mainly by volunteer, non-scientists, the data is thoroughly vetted and in my experience is high quality data.

The data analysis seems sound, although I did not dig deeply into how it was performed (I did not see where the code was included in the submission) and frankly, I lack the advanced R skills of co-authors Donovan and van Woesik. To be clear, the analytical framework seems solid but I cannot vouch for the under-the-hood details (I have neither the expertise nor the code).

The graphics are excellent (nice large font!), I love the coef plot, and the writing is clear and mostly concise. One exception: instead of "In the present study, the global correlation" I'd say "We found..." or similar. To me "In the present study" is so 18th century.

I have two concerns about the inferences made from the analysis. The main and strongest (but still quite mild) is about the latitudinal pattern of bleaching (less in topical areas) and the inference that that could be caused by coral diversity. First, I'm assuming that you saw a greater sensitivity to temperature at higher latitude, even while holding geographic differences in thermal characteristics constant? Second, given that coral diversity also varies very strongly with longitude, I think the authors need to formally include diversity as a covariate in the model. Charlie Veron has a shape file

based on his coral species range maps that we used in Zhang et al <https://peerj.com/articles/308/>
Ping me (jbruno@unc.edu) if you can't get it from Charlie. Without doing so, IMO the wording on the inferences about the effect of diversity is variable: sometimes OK, sometimes too strong.

This, IMO, is way too strong: in general equatorial reefs with high diversity are faring better than elsewhere." Mainly because the study measured bleaching frequency, not coral mortality from bleaching, coral loss, or current coral cover. I don't think based on the latter two variables, high diversity and / or tropical reefs are faring any better. In fact, its some of the highest latitude reefs doing the best.

And I don't see how the results support this inference:

"Our results do not necessarily suggest that high coral diversity
93 protects reefs from thermal stress, but rather that equatorial populations may support high
94 genotypic diversity that includes genotypes more tolerant of thermal stress."

It's a valid, testable hypothesis or explanation, but not a result, i.e., the results do not suggest equatorial populations may support high..."

Likewise:

"We demonstrated that equatorial areas and areas with greater exposure to SST
144 fluctuations may be more resilient to high temperature events, and therefore may be important
145 targets for conservation given their increased likelihood of persisting into the future."

The study didn't test whether tropical reefs were more resilient to high SST. You need to track coral cover before and after events to do this (which you could do...). And I know its tempting (a necessity?) to have this kind of conservation policy prescription in the conclusions, but really, can y'all not go there?

The other, very minor, concern is about the fact that coral bleaching sensitivity is declining. The question is why is this happening. My guess: selection for less thermally sensitive species, genera, and families (ie, not selection for tolerant genotypes of corals and zooxanthellae). Indirect evidence for this is the well-documented dependence of the effect of thermal anomalies on pre-disturbance coral cover (eg, doi: 10.1111/j.1365-2486.2012.02658.x) coupled with the observed shift in species composition (towards less-sensitive taxa).

"suggests that past
132 bleaching events may have culled the thermally susceptible individuals resulting in a recent
133 adjustment of the remaining coral populations to higher thresholds of bleaching
temperatures22."

Also, again be careful with the wording. The results don't suggest this mechanistic interpretation. That's the authors idea: it's reasonable, but it isn't a result. And I'd note other equally plausible alternatives.

Reviewer #3:

Remarks to the Author:

General comments

Mass coral bleaching events have occurred more frequently since the late 20th century due to increased levels of thermal stress as a result of global warming. The occurrence of bleaching often shows spatial variability and this study examines the relationship between bleaching patterns (based on Reef Check database) and a range of sea surface temperature (SST) metrics. The results confirm earlier studies that show bleaching is most common at sites with highest thermal stress and that there tends to be less bleaching at sites which experience high SST variance. New results arising from this study are that 1) geographically, bleaching is more likely to occur ~15-20° north or south of the equator compared to equatorial sites, and 2) that bleaching in the past decade occurred at SST ~ 0.5°C higher than in the preceding decade. This suggests that reefs have lost corals that are most sensitive to thermal stress and that the remaining populations are more thermally tolerant. This would be an important conclusion and of interest to the coral reef community and more widely. I am not, however, entirely convinced that the authors' findings fully support these potentially novel conclusions. There is a lack of clarity in the writing, nor are the most appropriate references cited in places. The potential limitations of the coral database, methods used and their justification also needs clarifying – at present it is very hard to follow. For example, the coral data base and analyses repeatedly refer to the period 1998-2017 yet the SST data base only appears to extend to 2012. There is also little discussion of other global-scale analyses of coral bleaching (e.g. Donner et al 2017; Oliver et al 2018). I provide below several specific comments which I feel the authors need to address before the manuscript is potentially suitable for publication. Even if these are addressed satisfactorily, I believe the study would be better targeted to a more specialised journal that allows a longer format, rather than the short format of Nature Communications.

Specific comments

Line 28: 'Recent mass coral bleaching.....' There are other causes of bleaching and it is only since the latter part of the 20th century that widespread bleaching due to thermal stress has been linked with climate change.

Lines 36-37: 'mid-latitude sites' could be misinterpreted as global mid-latitudes; suggest amend to 'tropical mid-latitude sites' or just 'sites 15-20°N or S of the equator'

Lines 47-48: coral bleaching does not cause the loss of the symbionts, rather it is the result of the coral's response to thermal stress that causes the loss of their symbionts resulting in coral bleaching.

Line 52: 'Most global models' – global models of what? Need to be more precise.

Lines 53-55: Need for greater precision, especially in relation to cited references. Hughes et al (2017, 2018 Refs # 2 & 10) do not cover bleaching in 2017; also these references do not support the statement about corals and other reef organisms being killed. These papers report the extent and intensity of bleaching and not coral mortality (see Hughes et al 2018 Nature doi:10.1038/s41586-018-0041-2 regarding coral mortality on the Great Barrier Reef, Australia after the 2016 bleaching event). Other references need to be provided to support the statement about death of 'other marine organisms'.

Lines 55-56: What sort of satellite data? I am not sure that Frieler et al (2012) is the best reference in support of the statement being made here.

Lines 56-58: This needs further explanation – unclear to me how 'local' variations in corals response to thermal stress are just a consequence of the daily, seasonal and inter-annual sea surface temperature (SST) regime. Also what about differing responses to thermal stress as a result of species, with some appearing to be more resistant than others (many references can be cited to support this).

Lines 58-62: Confusing – do the authors mean unusually warm SST rather than just 'high SST'? It is also unclear to me what the 'mismatch between global models and field results' exactly is – this needs to be explained more clearly.

Lines 63-66: Are the 3,351 sites individual reefs? If not, typically how many sites per reef? I presume each site record is continuous over the 20-year period. If not, then this should be noted. When referring to 'a range of environmental conditions' do the authors mean geographical variations in

average marine climate of the different sites or changing environmental conditions through time?

Lines 66-67: Need to provide reference and/or further explanation of why El Nino conditions are relevant – basically during typical El Nino events, large parts of the tropical oceans are warmer than usual which can increase the probability of thermal conditions conducive to bleaching.

Lines 67-69: Provide appropriate reference for the definition of DHWs – e.g. papers from NOAA's coral reef group.

Line 70: 'Our global model' needs to be described more fully. Global model of what? What type of model? What is the model predicting?

Line 70-71: I think the reasoning behind selecting these initial 30 temperature metrics needs to be more explicit – they just seem to be every possible metric that could be extracted from the SST data base and a bit more rationale is needed (briefly in the text and in more detail in the Supplementary Material).

Lines 74-77 and Figure 2: I feel the data sets used in these analyses are poorly described. It is unclear to me whether (see above) there is a continuous time series of bleaching 'prevalence' for each of the 3,351 sites – if so, what is the temporal resolution? I presume this is probably annual so how are these data compared to 'weekly' SST metrics (Table S1)? In the Figure 2 caption – time periods 1998-2017 and 1984-2017 are referred to – is this correct?

Lines 78-83 and Figure 3: The caption needs further explanation for people unfamiliar with this type of analysis. Basically (and I could be stupid), I do not understand what is being shown here, what the different colours mean and how it should be interpreted.

Lines 83-85 & Figures S1 and S2: These figures give 'frequency' which I presume to mean number of sites falling into each bleaching category by latitude and longitude, respectively. So is this frequency the number of sites in each category? If so, are these frequency plots scaled by the number of reef sites in each latitude or longitude category?

Line 86: As indicated earlier, use of the term 'mid-latitude' needs to be clarified as it will mean something else to many people.

Lines 86-88: Could some statistics be provided to support this statement – from Figure S1, more bleaching between 15-20N than at the equator seems to be true for the Northern Hemisphere but less so for the Southern Hemisphere? Also 'Figure 2' does not show bleaching vs latitude.

Lines 89-92: Do the authors mean 'biodiversity'? Is biodiversity relevant here? Unclear what point is being made?

Lines 95-110: Difficult to follow and not easily discerned from Figure 3b. Lines 106-107 – what does 'commonly experience fluctuations in SST' mean? Do the authors mean large daily, seasonal or inter-annual SST ranges? What are the cut-off values for these values, i.e. what is a large 'fluctuation' and what is a small 'fluctuation'.

Lines 111-112: Provide references to support statement regarding equatorial coral reefs having higher biodiversity.

Lines 112-115: Cited study appears to relate to tropics vs higher latitudes whereas the authors are comparing equatorial vs 15-20 degrees north/south of the equator. Confusing.

Lines 115-117: Need to define for the reader what is meant by 'climate velocities'.

Lines 124-125 & Figures 2 & 4: Unclear why Figure 2 is referenced here. Also Figure 4 caption says 'from 1998-2017' but the graph starts in 2002?

Lines 124-136 & Figure 5: Are the same reef sites being reported for the two decades, i.e. are apples being compared with apples? This needs to be made clear as if there are differences in the geographic location of the sites reporting bleaching in the two decades then this could be reflected in the SST.

Lines 154-160: I think a more detailed description of the Reef Check data base is required for readers unfamiliar with this resource. Some comments as to how it compares with other global coral bleaching data bases would also be useful, e.g. Donner et al (2017). Also some commentary about possible limitations of coral bleaching data sets would be useful, e.g. Oliver et al (2018).

Line 160: Table S1 does not provide a 'suite of ecological data' – just a description of the temperature metrics used.

Lines 165-166: Make it clear that these are 'coral' ecoregions. Also, Veron et al (2015) identify 150 coral ecoregions – please indicate how many coral ecoregions were used in this study.

Lines 167-170: Please indicate which version of CoRTAD was used. Also, the authors indicate that these temperature data cover the period 1982-2012. Yet, the analyses in the text are for the period 1998-2017 – this needs some explanation.

Line 174: Suggest using the term 'temperature metrics' here rather than 'covariates'. I also feel a bit more explanation is required (in simple language) as to what these different metrics describe and why they were chosen. Also Table S1 describes these as 'environmental parameters'. With the exception of 'depth', they are all related to temperature.

Lines 177-178: 'Sites were removed' – presumably these are the Reef Check coral survey sites? How many sites were removed?

The 1,435 page Supplementary Material needs to include descriptive captions for the data presented.

References:

Donner SD, GJM Rickbeil & SF Heron (2017) A new, high-resolution global mass coral bleaching database. PLoS ONE, doi:10.1371/journal.pone.0175490

Oliver JK, R Berkelmans & CM Eakin (2018) Coral bleaching in space and time. In: van Oppen MJH & JM Lough. Coral Bleaching. Patterns, Processes, Causes and Consequences. Springer, pp 27-49

Response to reviewers

Reviewer #1 (Remarks to the Author):

1. This paper analyses a strong and consistent dataset to make a clear point that bleaching levels are higher at mid-latitudes for coral reefs, and that the threshold for bleaching is significantly higher in the 2nd decade than the first of data collection. These are two very clear and important findings and should be published – they contribute to further science on understanding global change impacts, and help inform long term management planning.

Response: Thank you.

2. But the presentation of these results is muddled by mixing in two untested components – the relationship of these patterns with diversity of corals (no data presented) and higher reports of bleaching in the northern hemisphere (without addressing sampling artefacts that may influence this). I would have recommended minor revision to either remove or strengthen the references to diversity, but find after rereading that it is so mixed into the findings that the paper really needs a significant rewrite to do this well. Dealing with the geographic spread of sampling is also a significant addition to the conclusions from a dataset like this in a high-impact journal (and may, in fact, change the latitudinal pattern in some way). I believe resubmission is warranted, in a manuscript that focuses on the two defensible findings.

Response: We addressed both the diversity issue and sampling bias in the revised manuscript. We received coral diversity data from J.E.N. Veron, the world's leading coral taxonomists, and incorporated coral diversity in the generalized linear mixed models. The results (in Figure 2) show a highly significant negative relationship between coral bleaching and coral diversity, suggesting less coral bleaching occurs where coral diversity is high. We nevertheless are careful in our interpretation of these exciting results and suggest several hypotheses that could be causing the differential bleaching response in the revised manuscript.

To address the sampling bias, we added three new figures in the supplementary document (Figures S1, S2a, and S2b), and a new supplementary Table S1. These figures show the sampling effort, globally, and sampling sites categorized by latitude and longitude. The figures, especially Figure S2 shows that compared with elsewhere there were no differences in sampling effort at 15-20°N, where the majority of the coral bleaching occurred.

We also included a new table that shows the results of a series of Kolmogorov-Smirnov test results that compared bleaching frequencies at latitudes 15°–20° N and S with bleaching frequencies at other latitudes. The results in the table are clear. There was significantly more bleaching between the

latitudes 15 to 20 than elsewhere. We also ran a Spearman's correlation analysis to examine whether there was a relationship between coral bleaching and number of sites. There was no correlation ($\rho = 0.313$, $p\text{-value} = 0.297$), and these results are now reported in the revised manuscript.

We also ran a similar test to determine whether the northern and southern hemispheres differed in bleaching response. We found no significant difference in bleaching between the northern and southern hemispheres, and therefore we removed the one sentence that stated differences between hemispheres from the revised manuscript. Finally, the generalized linear mixed model, which we ran in a Bayesian framework, included sites as a *random effect*, which is the gold-star technique to remove sampling bias.

3. 42-43; 87-8- the paper does not address diversity, so it should not be mentioned, as is just correlative. It could easily – since species diversity for each of the Veron ecoregions is easily obtained. But if this is not added in, its more of a 'messaging' and subjective piece that is not backed up by any of the analysis in the paper.

Response: We have now addressed in the issue of diversity in the analysis (see Response 2 above), but remain cautious in the interpretation of the results.

4. 80 – this is six localities, not five

Response: Corrected.

5. 90-92- this feels a bit like cherry-picking a single reference, whereas there are many studies of regional patterns on bleaching in relation to latitude and diversity

Response: There are only a few studies that have rigorously tested the effect of diversity on coral bleaching, one of them was Heron et al. 2016, which we cite.

6. 93-94 – or that some other factor is related to lower bleaching at equatorial latitudes, that is unrelated to diversity. The paper does not really address diversity, and while the higher equatorial species diversity is implicit and generally known, it not a given that higher genotypic diversity is correlated with species diversity.

Response: We have now tested diversity explicitly (see also Response 2 above).

7. 111-112 “The results that coral bleaching was less common in the equatorial regions with highest diversity” ... equatorial regions have properties other than high diversity, such as the temperature gradients (velocities?) being lower than other regions. So the a priori association of genetic diversity as a reason explaining the patterns presented here is an assumption of the paper, and is not tested. It should

not be presented as a primary finding. Correlation, as implied here (and it is not even demonstrated statistically), does not mean causation.

Response: We have addressed diversity in the revised manuscript (see Response 2 above) but remained cautious in our interpretation.

8. 111-115 – latitudinal patterns of species diversity in corals are attributed to a wide range of Cenozoic influences (see Renema and others), rather than more recent glacial cycles, so even this citation of (terrestrial) diversity is not particularly accurate or relevant. There will be many citations for marine diversity patterns that would be more relevant, but may not indicate the same pattern.

Response: We removed Nolan et al. 2018 that referred to the terrestrial tropics, rewrote the sentence, and cited a marine-related study (Penn et al. 2018, *Science*).

9. 121-123 – “do not consider the role of species or genotypic diversity in driving the differences in thermal responses, or the potential of the genetic standing stock of corals to adapt to thermal stress” The role of these in the paper is speculation, but is presented as causation.

Response: We have toned down the boldness of the argument, although we explicitly added data on coral diversity to our analyses (see Response 2, above).

10. 125 – yes, ‘less predictable’, but nevertheless highly expected for 15-20 years now, so citations to this effect must be made, and include Hoegh-Gulberg 1999, Sheppard 2003, Hughes et al 2003 ...

Response: We agree and have included several of the mentioned citations.

11. 132-133 – and/or lead to acclimatization of surviving corals

Response: Agree. We have made the change.

12. 131-139 – this discussion on rates of change of SST should cite Donner 2009 (PLOS ONE) , who has estimated expected rates of change of temperature against adaptive capacity in coral communities. Reference #22 for this section seems a very narrow one to use, where more comprehensive syntheses have been made.

Response: Agree. We have cited the global study by Donner 2009.

13. 140-144 These sentences and logic are somewhat jumbled, flipping between explanations based on the study “results have important implications for improving predictions of future bleaching” and findings “We demonstrated that equatorial areas ...”, but without attributing the original ideas where they are due. This study confirms what is known about this topic with a new analysis of a

complete/updated dataset. It adds strength to one hypothesis, that survival is better at equatorial latitudes because rates of warming are less (compared to an alternate hypothesis that survival might be less at the equator as absolute temperatures will be higher) but it does not tell us anything we did not already know (and in fact, this alternate hypothesis is not stated explicitly). For example, the predictions of greater survival of corals near the equator is consistent with Beyer et al. 2018 (though this paper does not make a specific statement about this), but this is not mentioned or cited.

Response: We rewrote these sentences to clarify our intent, and included a reference to Beyer et al. 2018.

14. 170 – SST data only to 2012? What about later years?

Response: We apologize for the error here, the date should read 2017. We have corrected the date in the revised manuscript.

15. Figs S1 – the predominance of bleaching in northern latitudes may be an artefact of sampling effort at these latitudes, and less in the south. To really say bleaching is more prevalent in the north sampling effort in relation to reef area needs to be analyzed. It is also possible that the higher sampling levels in the north are strongly biased to a small number of popular locations for this method, so patterns may be an artefact of this.

Response: We addressed the issue of sampling bias in Response 2 above.

16. Methods – need to state more clearly which variates were obtained from field data, and which from the lat-long points of field sites extracted from CoRTAD data.

Response: We clarified the methods, elaborated on the selection of environmental variables and temperature metrics, clarified our criteria for inclusion in the analysis, and made explicit where we obtained the data in Tables S2 and S3.

Reviewer #2 (Remarks to the Author):

17. Overall, I think this is an excellent, important, and timely paper/analysis. I am familiar with both of the core databases (ReefCheck and CorTAD) and have published papers based on both. Although the ReefCheck data is collected mainly by volunteer, non-scientists, the data is thoroughly vetted and in my experience is high quality data.

Response: Thank you.

18. The data analysis seems sound, although I did not dig deeply into how it was performed (I did not see where the code was included in the submission) and frankly, I lack the advanced R skills of co-authors Donovan and van Woeseik. To be clear, the analytical framework seems solid but I cannot vouch for the under-the-hood details (I have neither the expertise nor the code).

Response: Thank you. For reproducibility we have made all the R and OpenBugs code available at: <https://github.com/InstituteForGlobalEcology>.

19. The graphics are excellent (nice large font!), I love the coef plot, and the writing is clear and mostly concise. One exception: instead of "In the present study, the global correlation" I'd say "We found..." or similar. To me "In the present study" is so 18th century.

Response: Agree. We changed the text to "We found..."

20. I have two concerns about the inferences made from the analysis. The main and strongest (but still quite mild) is about the latitudinal pattern of bleaching (less in topical areas) and the inference that that could be caused by coral diversity. First, I'm assuming that you saw a greater sensitivity to temperature at higher latitude, even while holding geographic differences in thermal characteristics constant? Second, given that coral diversity also varies very strongly with longitude, I think the authors need to formally include diversity as a covariate in the model. Charlie Veron has a shape file based on his coral species range maps that we used in Zhang et al <https://peerj.com/articles/308/> Ping me (jbruno@unc.edu) if you can't get it from Charlie. Without doing so, IMO the wording on the inferences about the effect of diversity is variable: sometimes OK, sometimes too strong.

Response: We have addressed diversity in the revised manuscript (see Response 2, above).

21. This, IMO, is way too strong: in general equatorial reefs with high diversity are faring better than elsewhere." Mainly because the study measured bleaching frequency, not coral mortality from bleaching, coral loss, or current coral cover. I don't think based on the latter two variables, high diversity and / or tropical reefs are faring any better. In fact, its some of the highest latitude reefs doing the best.

Response: Thank you for the comment. In the revised manuscript, we show a significant negative relationship between coral bleaching and coral diversity. We do however tone down the rhetoric of causation and suggest several alternative hypotheses (see Response 2).

22. And I don't see how the results support this inference:

"Our results do not necessarily suggest that high coral diversity

93 protects reefs from thermal stress, but rather that equatorial populations may support high

94 genotypic diversity that includes genotypes more tolerant of thermal stress."

It's a valid, testable hypothesis or explanation, but not a result, i.e., the results do not suggest equatorial populations may support high..."

Likewise:

"We demonstrated that equatorial areas and areas with greater exposure to SST
144 fluctuations may be more resilient to high temperature events, and therefore may be important
145 targets for conservation given their increased likelihood of persisting into the future."
The study didn't test whether tropical reefs were more resilient to high SST. You need to track coral
cover before and after events to do this (which you could do...). And I know its tempting (a necessity?)
to have this kind of conservation policy prescription in the conclusions, but really, can y'all not go there?

Response: We tested coral diversity in the revised manuscript (see Response 2,
and 22), and toned down the language and suggested several alternative
hypotheses.

23. The other, very minor, concern is about the fact that coral bleaching sensitivity is declining. The
question is why is this happening. My guess: selection for less thermally sensitive species, genera, and
families (ie, not selection for tolerant genotypes of corals and zooxanthellae). Indirect evidence for this
is the well-documented dependence of the effect of thermal anomalies on pre-disturbance coral cover
(eg, doi: 10.1111/j.1365-2486.2012.02658.x) coupled with the observed shift in species composition
(towards less-sensitive taxa).

"suggests that past
132 bleaching events may have culled the thermally susceptible individuals resulting in a recent
133 adjustment of the remaining coral populations to higher thresholds of bleaching temperatures²²."
Also, again be careful with the wording. The results don't suggest this mechanistic interpretation. That's
the authors idea: it's reasonable, but it isn't a result. And I'd note other equally plausible alternatives.

Response: Thank you for the comment. There is a subtle difference between
selection for less sensitive species to thermal stress and selection for more
thermally tolerant species. We however changed the wording to "less sensitive to
thermal stress" in the revised text, and suggested several hypotheses as to
potential mechanisms (see also Response 21).

Reviewer #3 (Remarks to the Author):

24. General comments

Mass coral bleaching events have occurred more frequently since the late 20th century due to increased
levels of thermal stress as a result of global warming. The occurrence of bleaching often shows spatial
variability and this study examines the relationship between bleaching patterns (based on Reef Check
database) and a range of sea surface temperature (SST) metrics. The results confirm earlier studies that
show bleaching is most common at sites with highest thermal stress and that there tends to be less
bleaching at sites which experience high SST variance. New results arising from this study are that 1)
geographically, bleaching is more likely to occur ~15-20o north or south of the equator compared to
equatorial sites, and 2) that bleaching in the past decade occurred at SST ~ 0.5oC higher than in the
preceding decade. This suggests that reefs have lost corals that are most sensitive to thermal stress and

that the remaining populations are more thermally tolerant. This would be an important conclusion and of interest to the coral reef community and more widely. I am not, however, entirely convinced that the authors' findings fully support these potentially novel conclusions. There is a lack of clarity in the writing, nor are the most appropriate references cited in places.

Response: We have clarified the issues that were raised by this reviewer with regard to the writing, and have changed, where appropriate, some of the references.

25. The potential limitations of the coral database, methods used and their justification also needs clarifying – at present it is very hard to follow. For example, the coral data base and analyses repeatedly refer to the period 1998-2017 yet the SST data base only appears to extend to 2012. There is also little discussion of other global-scale analyses of coral bleaching (e.g. Donner et al 2017; Oliver et al 2018). I provide below several specific comments which I feel the authors need to address before the manuscript is potentially suitable for publication. Even if these are addressed satisfactorily, I believe the study would be better targeted to a more specialised journal that allows a longer format, rather than the short format of Nature Communications.

Response: We have clarified the methods and justified the selection of the environmental variables. The database extends to 2017 (not to 2012 as we mistakenly had written in the initial manuscript). We have also included the suggested citations made by the reviewer. We thank the reviewer for the suggestions and comments.

26. Specific comments

Line 28: 'Recent mass coral bleaching.....' There are other causes of bleaching and it is only since the latter part of the 20th century that widespread bleaching due to thermal stress has been linked with climate change.

Response: The reviewer is correct, bleaching also can be caused by changes in salinity and other stressors, therefore to be more specific, and without going into too much detail in the opening sentence, the sentence has been changed to:

"Recent coral bleaching and the subsequent dramatic loss in coral cover are caused by thermal-stress events associated with climate change¹⁻²".

27. Lines 36-37: 'mid-latitude sites' could be misinterpreted as global mid-latitudes; suggest amend to 'tropical mid-latitude sites' or just 'sites 15-20oN or S of the equator'

Response: Amended. Thank you.

28. Lines 47-48: coral bleaching does not cause the loss of the symbionts, rather it is the result of the coral's response to thermal stress that causes the loss of their symbionts resulting in coral bleaching.

Response: The sentence has been rewritten to reflect that loss of symbionts is caused by thermal stress.

29. Line 52: 'Most global models' – global models of what? Need to be more precise.

Response: We have been more specific and stated: *“Most studies that examine coral response to coarse-grained global circulation models predict that within the next eighty years few coral reefs will survive in tropical oceans⁹”*.

30. Lines 53-55: Need for greater precision, especially in relation to cited references. Hughes et al (2017, 2018 Refs # 2 & 10) do not cover bleaching in 2017; also these references do not support the statement about corals and other reef organisms being killed. These papers report the extent and intensity of bleaching and not coral mortality (see Hughes et al 2018 Nature doi:10.1038/s41586-018-0041-2 regarding coral mortality on the Great Barrier Reef, Australia after the 2016 bleaching event). Other references need to be provided to support the statement about death of 'other marine organisms'.

Response: We have changed the referencing to 8,10, (Stuart-Smith et al. 2018; and Hughes et al. 2018), both papers reported bleaching and mortality.

31. Lines 55-56: What sort of satellite data? I am not sure that Frieler et al (2012) is the best reference in support of the statement being made here.

Response: We removed the citation to Frieler et al. 2012 in this sentence.

32. Lines 56-58: This needs further explanation – unclear to me how 'local' variations in corals response to thermal stress are just a consequence of the daily, seasonal and inter-annual sea surface temperature (SST) regime. Also what about differing responses to thermal stress as a result of species, with some appearing to be more resistant than others (many references can be cited to support this).

Response: We tried to cram too much into the sentence. Indeed, many studies, including many of our own publications, show species-specific responses to thermal stress. We have rewritten the sentence and clarified our intent.

33. Lines 58-62: Confusing – do the authors mean unusually warm SST rather than just 'high SST'? It is also unclear to me what the 'mismatch between global models and field results' exactly is – this needs to be explained more clearly.

Response: We changed the text to read anomalously high temperatures, instead of just high temperatures. The mismatch is more clearly explained in the revised text: *“Compared with coarse-grained global models that predict minimal coral survival in the tropical oceans within the next one-hundred years, recent field work shows considerable geographic variability in both temperature stress and coral survival¹¹⁻¹⁴. This mismatch...”*

34. Lines 63-66: Are the 3,351 sites individual reefs? If not, typically how many sites per reef? I presume each site record is continuous over the 20-year period. If not, then this should be noted. When referring to ‘a range of environmental conditions’ do the authors mean geographical variations in average marine climate of the different sites or changing environmental conditions through time?

Response: We added more information on the sampling in the methods section. The sites are reefs, but each reef was not sampled every year. We have been more specific in the methods section about characterizing the environmental variables, since the environmental variables were the changing environmental conditions through time.

35. Lines 66-67: Need to provide reference and/or further explanation of why El Nino conditions are relevant – basically during typical El Nino events, large parts of the tropical oceans are warmer than usual which can increase the probability of thermal conditions conducive to bleaching.

Response: Thank you. The inclusion of the term El Niño conditions has now been justified in this sentence.

36. Lines 67-69: Provide appropriate reference for the definition of DHWs – e.g. papers from NOAA’s coral reef group.

Response: Appropriate citation has now been included for Degree Heating Weeks (Gleason and Strong 1996).

37. Line 70: ‘Our global model’ needs to be described more fully. Global model of what? What type of model? What is the model predicting?

Response: We have been more specific and wrote that our model predicts coral bleaching.

38. Line 70-71: I think the reasoning behind selecting these initial 30 temperature metrics needs to be more explicit – they just seem to be every possible metric that could be extracted from the SST data base and a bit more rationale is needed (briefly in the text and in more detail in the Supplementary Material).

Response: We have gone into more detail to rationalize the selective process of the environmental variables, in both the methods section and in the supplementary document.

39. Lines 74-77 and Figure 2: I feel the data sets used in these analyses are poorly described. It is unclear to me whether (see above) there is a continuous time series of bleaching ‘prevalence’ for each of the 3,351 sites – if so, what is the temporal resolution? I presume this is probably annual so how are these data compared to ‘weekly’ SST metrics (Table S1)? In the Figure 2 caption – time periods 1998-2017 and 1984-2017 are referred to – is this correct?

Response: See also Response 35. We have revisited the explanations of the datasets and have refined our explanations in the revised manuscript. The environmental metrics were extracted at the time of field sampling for coral bleaching, and, as stated in Response 35, the environmental variables were measured as the changing environmental conditions through time. The revised manuscript reads: *“the coral community bleaching response was recorded using the same standardized protocol at each site across a suite of changing environmental variables from 1998 through to 2017....”*

We have further clarified text relating to the sampling frequency in the revised manuscript, and have provided a new supplementary figure that highlights sampling effort (Figure S1).

40. Lines 78-83 and Figure 3: The caption needs further explanation for people unfamiliar with this type of analysis. Basically (and I could be stupid), I do not understand what is being shown here, what the different colours mean and how it should be interpreted.

Response: We clarified the text and the figure caption. The figure essentially captures the relationship of coral bleaching with the frequency of thermal-stress anomalies across the oceans. The red colors convey a more positive relationship and the blue colors convey a more negative relationship. The stronger the colors the stronger the relationships, or more technically the steeper the slope of B_2 in equation 7. All the code to produce the figures is deposited at <https://github.com/InstituteForGlobalEcology>.

41. Lines 83-85 & Figures S1 and S2: These figures give ‘frequency’ which I presume to mean number of sites falling into each bleaching category by latitude and longitude, respectively. So is this frequency the number of sites in each category? If so, are these frequency plots scaled by the number of reef sites in each latitude or longitude category?

Response: The frequency refers to the number of surveys, not the number of sites falling into bleaching by latitude and longitude. A single site may have multiple studies over the entire time frame. The figures provide information on the frequency of bleaching and not bleaching, which together provides information

on the sampling effort. We have also made a new figure which shows the number of sites at each latitude and longitude. The figure captions have been clarified.

42. Line 86: As indicated earlier, use of the term ‘mid-latitude’ needs to be clarified as it will mean something else to many people.

Response: We have changed the term to tropical mid-latitudes.

43. Lines 86-88: Could some statistics be provided to support this statement – from Figure S1, more bleaching between 15-20N than at the equator seems to be true for the Northern Hemisphere but less so for the Southern Hemisphere? Also ‘Figure 2’ does not show bleaching vs latitude.

Response: We have undertaken some statistical analyses (i.e., a series of Kolmogorov-Smirnov tests) which show that the latitudes 15-20 had significantly more bleaching than elsewhere (Table S1). Figure 2 is still relevant because it shows that latitude played a role in coral bleaching, and Figure S2 shows that there is more bleaching at the suggested latitudes.

44. Lines 89-92: Do the authors mean ‘biodiversity’? Is biodiversity relevant here? Unclear what point is being made?

Response: We have clarified this term. We meant coral diversity, rather than diversity, or “biodiversity” as was suggested by the reviewer. We included a citation to coral diversity in the revised manuscript (see also Response 2).

45. Lines 95-110: Difficult to follow and not easily discerned from Figure 3b. Lines 106-107 – what does ‘commonly experience fluctuations in SST’ mean? Do the authors mean large daily, seasonal or inter-annual SST ranges? What are the cut-off values for these values, i.e. what is a large ‘fluctuation’ and what is a small ‘fluctuation’.

Response: We clarified this ambiguity as: “localities that commonly experience large daily or seasonal SST ranges may harbor corals...” . We do not provide a cut-off, or a threshold, because such fluctuations are a continuous variable, and as shown in Figure 3 there is a range of responses.

46. Lines 111-112: Provide references to support statement regarding equatorial coral reefs having higher biodiversity.

Response: We provided a reference to Veron (2000) Corals of World.

47. Lines 112-115: Cited study appears to relate to tropics vs higher latitudes whereas the authors are comparing equatorial vs 15-20 degrees north/south of the equator. Confusing.

Response: We have rethought and rewrote this sentence and included different references (see also Response 8).

48. Lines 115-117: Need to define for the reader what is meant by 'climate velocities'.

Response: We have defined climate velocities in the revised manuscript as the rate and direction that the climate shifts across the seascape.

49. Lines 124-125 & Figures 2 & 4: Unclear why Figure 2 is referenced here. Also Figure 4 caption says 'from 1998-2017' but the graph starts in 2002?

Response: Figure 4 is much clearer, and therefore we removed the reference to Figure 2. We have corrected the starting date of the timeline.

50. Lines 124-136 & Figure 5: Are the same reef sites being reported for the two decades, i.e. are apples being compared with apples? This needs to be made clear as if there are differences in the geographic location of the sites reporting bleaching in the two decades then this could be reflected in the SST.

Response: Yes, the same sites are being reported through time. This has been clarified in the revised manuscript.

51. Lines 154-160: I think a more detailed description of the Reef Check data base is required for readers unfamiliar with this resource. Some comments as to how it compares with other global coral bleaching data bases would also be useful, e.g. Donner et al (2017). Also some commentary about possible limitations of coral bleaching data sets would be useful, e.g. Oliver et al (2018).

Response: We have added some more details on the data in the methods section; we do not purposefully compare our data to others, especially to Donner et al. (2017), to which incidentally van Woesik contributed, because the Donner dataset does not contain information on when bleaching was absent. It only reports positive bleaching records. We used the Donner data in some preliminary data analyses and soon realized that making predictions without absences was problematic and the models had extremely high uncertainty (i.e., Bayesian credible intervals). We avoided such discussions in the present manuscript.

52. Line 160: Table S1 does not provide a ‘suite of ecological data’ – just a description of the temperature metrics used.

Response: We rephrased the Table caption.

53. Lines 165-166: Make it clear that these are ‘coral’ ecoregions. Also, Veron et al (2015) identify 150 coral ecoregions – please indicate how many coral ecoregions were used in this study.

Response: We added the word coral to the text, which now reads: “coral ecoregions”. We have also indicated how many coral ecoregions (77) were used in the study in Table S3.

54. Lines 167-170: Please indicate which version of CoRTAD was used. Also, the authors indicate that these temperature data cover the period 1982-2012. Yet, the analyses in the text are for the period 1998-2017 – this needs some explanation.

Response: In the revised manuscript we have indicated that we used CoRTAD Version 6, which continues until 2017. We have corrected the one instance where we incorrectly stated 1982-2012 instead of 1982-2017.

55. Line 174: Suggest using the term ‘temperature metrics’ here rather than ‘covariates’. I also feel a bit more explanation is required (in simple language) as to what these different metrics describe and why they were chosen. Also Table S1 describes these as ‘environmental parameters’. With the exception of ‘depth’, they are all related to temperature.

Response: We agree that most of the variables are temperature metrics, however we also used depth, latitude, year, and diversity, which are not temperature metrics. Therefore, technically it is more correct to use the term covariate to encompass temperature metrics and other variables (see also Response 17 regarding the selection of the variables).

56. Lines 177-178: ‘Sites were removed’ – presumably these are the Reef Check coral survey sites? How many sites were removed?

Response: Yes they were Reef Check sites. We removed 153 sites (4%), which is stated in the revised methods section, and in the table caption of Table S3).

57. The 1,435 page Supplementary Material needs to include descriptive captions for the data presented.

Response: We have included descriptive captions for the data presented in the

supplementary document, and now have all the code to reproduce the figures in annotated code available at: <https://github.com/InstituteForGlobalEcology>

Again, our thanks extend to the three reviewers whose efforts have improved the manuscript. Their efforts have been acknowledged in the revised acknowledgement section.

Reviewers' Comments:

Reviewer #1:

Remarks to the Author:

Line 36 the conclusion of higher bleaching "for locations with higher diversity" has been elevated in prominence here, being introduced before and separately from the caveat that diversity is higher at lower latitudes, and lower latitude reefs show less bleaching than mid-latitude reefs because related to temperature dynamics. Far from "remaining cautious in the interpretation of the results" the abstract goes the opposite direction from both reviews.

I believe this comes from adding diversity into the regression, which does in fact strengthen the result that bleaching is inversely related to diversity – but rather than resolving the discussion on mechanism, that the reviews identified, only strengthens what may be a spurious correlation. I'm not a statistician, but I believe the source of this comes from the unequal sampling among ecoregions, and if I'm not mistaken, the same species diversity figure is introduced into the regression for all sites within an ecoregion. Thus diversity behaves more as a factor (with 75 levels as that's the number of ecoregions with data) than a variable (that should be continuous, for the 3351 cases in the study). That is, 12% of all cases have the same diversity for the Sunda shelf (low latitude, low bleaching), 8% for SE Asia/Philippines (low latitude, low bleaching) and Hispaniola (mid-latitude, high bleaching) each, etc etc. This only strengthens my interpretation that the finding on bleaching and diversity patterns is biased by sampling.

So I think important changes to the abstract need to be along the lines of:

Lines 34-36 – "However, coral bleaching was significantly lower in localities with a high variance in sea-surface temperature (SST) anomalies and in localities with high coral diversity" should be altered towards something like this "However, coral bleaching was significantly lower in localities with a high variance in sea-surface temperature (SST) anomalies which were predominantly in low latitude ecoregions with high coral diversity

And

Lines 42-44 – "Although persistent thermal-stress events are bleaching reef corals worldwide, in general equatorial reefs with high diversity are faring better than elsewhere" to something like "Although persistent thermal-stress events are bleaching reef corals worldwide, in general equatorial reefs where coral diversity is highest are faring better than elsewhere"

Because no track-changes document is provided it's too time-consuming to do a sentence-by-sentence and section-by-section review of what changes the authors have made, in relation to their rebuttal letter, so here I'm just indicating some sections that I still have discomfort with:

89- "and at sites with low coral diversity" – this suggests that within an ecoregion, bleaching was higher at sites with low diversity compared to sites with high diversity. Whereas, the comparison is sites in ecoregions with low diversity vs in ecoregions with high diversity.

98-104 – I find that the emphasis on diversity (and genotypes – where no justification is provided for the presence of higher genotype diversity in low latitudes, just as in the first manuscript, little evidence was presented on species diversity) is still cherry-picking. An additional hypothesis includes the real differences that are stated in lines 80-82 (localities with high SST, DHW, frequently high SST

anomalies, rate of change in SST), in that these are latitudinally correlated (fig. S16).

A couple of weaknesses in Response 2:

1) it leaves out presenting the relationship between diversity and other variables, diversity is not in fig. S16, and its relation to S17 (bleaching vs. latitude) and S21 (bleaching vs SST temp anom).

2) In relation to my point about diversity as a factor not variable - "We also ran a Spearman's correlation analysis to examine whether there was a relationship between coral bleaching and number of sites. There was no correlation ($\rho = 0.313$, $p\text{-value} = 0.297$), and these results are now reported in the revised manuscript." The table below shows a very quick summary combining the visual results in fig. S1 (latitude and number of surveys) with Table S3 (ecoregion and number of surveys) but with the missing information (see point 1 above) about the degree of bleaching in these ecoregions (in fig. 1 but for points, not for ecoregions) ... I'm not sure where this is going but it suggests a hump-shaped curve rather than the linear one that Spearman tests for, so I'm left with more questions than answers.

# ecoregion	#surveys	%surveys	Cum%	Latitude	bleaching	
49 Sunda Shelf, south-east Asia	929	12%	12%	low	??	
46 South-east Philippines	632	632	8%	20%	low	??
138 Hispaniola, Puerto Rico and	582	8%	28%	mid	??	
45 Sulu Sea	579	579	7%	35%	low	??
115 Society Islands, French Polynesia	516	7%	42%	mid	??	
136 Belize and west Caribbean	420	5%	47%	mid	??	
78 Central and northern Great	378	5%	52%	mid	??	
55 Hong Kong	278	278	4%	56%	high	??

Conclusion – I'm apologetic that I don't have the time to review this revision in full – the lack of a Track Changes version makes it too unwieldy for all the nuances that are important to track. I'm not convinced by the primal importance of the diversity correlation, and while both reviews have raised this as a primary question, the authors have in fact elevated this finding in the abstract. My own feeling is that the main two findings, which are important (see first review) should be emphasized, and the pattern in diversity mentioned, but not presented as a potential causal factor. The format in this paper is too short to do it meaningfully, and what is there detracts from the main findings. A second paper!! With or without these revisions, I leave the next and final decisions to the editor.

Reviewer #2:

Remarks to the Author:

The authors have carefully and thoroughly responded to the reviewer comments and suggestions to improve the manuscript. They obtained data on coral species richness, and included richness as a cofactor in the model. The result supports their original interpretation. They also modified the text and toned-down some of the arguments that the reviewers felt were not supported by the analysis. Overall, the ms is now, in my opinion, ready for acceptance and publication without further review or modification.

Reviewer #3:

Remarks to the Author:

Having read the revised manuscript and the detailed responses to the comments of the three referees, I am satisfied that the authors have appropriately revised the manuscript and that it is now suitable for publication.

Thank you for the second round of reviewers' comments on our manuscript: Coral bleaching: A global analysis.

We appreciate the positive responses by Reviewers 2 and 3 and have addressed all seven comments made by Reviewer 1 below. We address the concerns about sampling bias made by Reviewer 1 and have modified our model to now include a hierarchical effect of ecoregion, since ecoregions with more samples are more certain than those with less samples. We also have backed off from the suggestion that coral diversity is a cause of less bleaching given the results of the new model, which suggests no effect of diversity on bleaching. We have removed or reworded any text that makes mention of diversity directly reducing bleaching, and propose three testable hypotheses that address the issue of lower bleaching in the low-latitude tropics.

Reviewer #1 (Remarks to the Author):

1) Line 36 the conclusion of higher bleaching “for locations with higher diversity” has been elevated in prominence here, being introduced before and separately from the caveat that diversity is higher at lower latitudes, and lower latitude reefs show less bleaching than mid-latitude reefs because related to temperature dynamics. Far from “remaining cautious in the interpretation of the results” the abstract goes the opposite direction from both reviews. I believe this comes from adding diversity into the regression, which does in fact strengthen the result that bleaching is inversely related to diversity – but rather than resolving the discussion on mechanism, that the reviews identified, only strengthens what may be a spurious correlation. I’m not a statistician, but I believe the source of this comes from the unequal sampling among ecoregions, and if I’m not mistaken, the same species diversity figure is introduced into the regression for all sites within an ecoregion. Thus diversity behaves more as a factor (with 75 levels as that’s the number of ecoregions with data) than a variable (that should be continuous, for the 3351 cases in the study). That is, 12% of all cases have the same diversity for the Sunda shelf (low latitude, low bleaching), 8% for SE Asia/Philippines (low latitude, low bleaching) and Hispaniola (mid-latitude, high bleaching) each, etc etc. This only strengthens my interpretation that the finding on bleaching and diversity patterns is biased by sampling. So I think important changes to the abstract need to be along the lines of: Lines 34-36 – “However, coral bleaching was significantly lower in localities with a high variance in sea-surface temperature (SST) anomalies and in localities with high coral diversity” should be altered towards something like this “However, coral bleaching was significantly lower in localities with a high variance in sea-surface temperature (SST) anomalies which were predominantly in low latitude ecoregions with high coral diversity

Response: The reviewer is correct that diversity was introduced at the ecoregion level. Given the concerns over introducing eco-region level diversity at the site level, we have modified our model to now include a hierarchical effect of ecoregion on the intercept. We then modeled diversity as a function of the ecoregion means. This approach also addresses the reviewers concerns about sampling bias because relationships can deviate by ecoregion, and ecoregions with more samples are more certain than those with less. To further address this comment, as mentioned above, we added additional cautionary language to our

interpretation of diversity throughout the revised text (lines 88–89, and 99–100) and have made the suggested changes to the Abstract (lines 36–37 and 44–46).

2. And Lines 42-44 – “Although persistent thermal-stress events are bleaching reef corals worldwide, in general equatorial reefs with high diversity are faring better than elsewhere” to something like “Although persistent thermal-stress events are bleaching reef corals worldwide, in general equatorial reefs where coral diversity is highest are faring better than elsewhere”

Response: The change was made to the Abstract as suggested.

3. Because no track-changes document is provided it's too time-consuming to do a sentence-by-sentence and section-by-section review of what changes the authors have made, in relation to their rebuttal letter, so here I'm just indicating some sections that I still have discomfort with:

89- “and at sites with low coral diversity” – this suggests that within an ecoregion, bleaching was higher at sites with low diversity compared to sites with high diversity. Whereas, the comparison is sites in ecoregions with low diversity vs in ecoregions with high diversity.

Response: We removed the reference to sites and exchanged the word ecoregions. The text now reads: "than in the equatorial regions, where coral diversity is highest."

4. 98-104 – I find that the emphasis on diversity (and genotypes – where no justification is provided for the presence of higher genotype diversity in low latitudes, just as in the first manuscript, little evidence was presented on species diversity) is still cherry-picking. An additional hypothesis includes the real differences that are stated in lines 80-82 (localities with high SST, DHW, frequently high SST anomalies, rate of change in SST), in that these are latitudinally correlated (fig. S16).

Response: We provided 3 testable hypotheses in the original manuscript, and now have added an additional hypothesis in the revised text that coral bleaching was simply a consequence of reduced thermal stress.

The revised text reads: *Unless there was less thermal stress in the low-latitude tropics than elsewhere, which we did not detect in this study, our results lead to several hypotheses that potentially explain differential coral bleaching among latitudes. We hypothesize that the low-latitude tropics bleached less because: (i) of the geographical differences in species composition, (ii) of the higher genotypic diversity at low latitudes, which include genotypes less susceptible to thermal stress, (iii) some corals were preadapted to thermal stress because of consistently warmer temperatures at low latitude. These hypotheses are not mutually exclusive and several of these mechanisms could be operating in concert, resulting in less coral bleaching at low latitudes.*

5. A couple of weaknesses in Response 2:

it leaves out presenting the relationship between diversity and other variables, diversity is not in fig. S16, and its relation to S17 (bleaching vs. latitude) and S21 (bleaching vs SST temp anom).

Response: We aren't 100% positive what the reviewer is asking for here, but it seems that the reviewer is suggesting that we compare coral diversity with "other [environmental] variables". We did not include diversity in S16 (correlations among predictors) given that we only have 1 value for each ecoregion. Otherwise, perhaps the reviewer is alluding to interactions among predictors, and if so, we did not include any interactive terms in the model given the large number of predictors we were testing. Nonetheless, diversity is now included at the ecoregion level, so we are unable to test for interactions with other variables given the different scales of inference.

6) In relation to my point about diversity as a factor not variable - "We also ran a Spearman's correlation analysis to examine whether there was a relationship between coral bleaching and number of sites. There was no correlation ($\rho = 0.313$, $p\text{-value} = 0.297$), and these results are now reported in the revised manuscript." The table below shows a very quick summary combining the visual results in fig. S1 (latitude and number of surveys) with Table S3 (ecoregion and number of surveys) but with the missing information (see point 1 above) about the degree of bleaching in these ecoregions (in fig. 1 but for points, not for ecoregions) ... I'm not sure where this is going but it suggests a hump-shaped curve rather than the linear one that Spearman tests for, so I'm left with more questions than answers.

# ecoregion	#surveys	%surveys	Cum%	Latitude	bleaching	
49	Sunda Shelf, south-east Asia	929	12%	12%	low ??	
46	South-east Philippines	632	632	8%	20%	low ??
138	Hispaniola, Puerto Rico and	582	8%	28%	mid ??	
45	Sulu Sea	579	579	7%	35%	low ??
115	Society Islands, French Polynesia	516	7%	42%	mid ??	
136	Belize and west Caribbean	420	5%	47%	mid ??	
78	Central and northern Great	378	5%	52%	mid ??	
55	Hong Kong	278	278	4%	56%	high ??

Response: It is unclear which data are humped in the 5 variables that the reviewer describes. It seems that the reviewer is referring to a potential humped-shaped curve that may be attributed to an unequal number of sites relative to latitude, that were impacted by low, mid, and high bleaching. It seems that reviewer is interchangeably using "site" and "survey", whereas they are different entities (we have 9000+ surveys and 3000+sites). Figures S1 and S3 show that there are more surveys between latitudes 15 and 20 degrees, although there are no proportional differences between moderate and severe bleaching and the number of surveys with no bleaching across latitudes.

7. Conclusion – I'm apologetic that I don't have the time to review this revision in full – the lack of a Track Changes version makes it too unwieldy for all the nuances that are important to track. I'm not convinced by the primal

importance of the diversity correlation, and while both reviews have raised this as a primary question, the authors have in fact elevated this finding in the abstract. My own feeling is that the main two findings, which are important (see first review) should be emphasized, and the pattern in diversity mentioned, but not presented as a potential causal factor. The format in this paper is too short to do it meaningfully, and what is there detracts from the main findings. A second paper!! With or without these revisions, I leave the next and final decisions to the editor.

Response: We thank the reviewer for the comments on diversity and agree that the other two main findings in the manuscript (i.e., latitudinal trends and bleaching at higher temperatures in the last decade) are substantial enough. Therefore, we were cautious in our interpretation of the relationship between coral bleaching and diversity, and focus on the geographic trends, in that low latitude sites bleach less than elsewhere.

Please note we have also moved Figure 3 to the supplementary document. The figure was peripherally associated with the main trust of the manuscript, in fact only encompassing one sentence. Although we still refer to the figure, we considered it more appropriate to place the figure in the supplementary document.

Thank you again for the suggestions and comments.